# Role of Single Nucleotide Variants in the *YAP1* Gene in Adolescents with Polycystic Ovary Syndrome

**DOI:** 10.3390/biomedicines10071688

**Published:** 2022-07-13

**Authors:** Lasma Lidaka, Laine Bekere, Gunta Lazdane, Marija Lazovska, Iveta Dzivite-Krisane, Linda Gailite

**Affiliations:** 1Department of Paediatric Gynaecology, Children’s Clinical University Hospital, LV-1004 Riga, Latvia; 2Department of Obstetrics and Gynaecology, Riga Stradins University, LV-1007 Riga, Latvia; gunta.lazdane@rsu.lv; 3Faculty of Residency, Riga Stradins University, LV-1007 Riga, Latvia; laine.bekere@gmail.com; 4Scientific Laboratory of Molecular Genetics, Riga Stradins University, LV-1007 Riga, Latvia; marija.lazovska@rsu.lv (M.L.); linda.gailite@rsu.lv (L.G.); 5Department of Paediatric Endocrinology, Children’s Clinical University Hospital, LV-1004 Riga, Latvia; dzivite@bkus.lv

**Keywords:** PCOS, adolescent, *YAP1* gene

## Abstract

Background: Polycystic ovary syndrome (PCOS) is one of the most common endocrinopathies in women. It can manifest in adolescence, affecting up to 8% of adolescents. Long-term health consequences characteristic of PCOS are impaired fertility, increased risk of type 2 diabetes, metabolic disorders and cardiovascular disease. All of these sequelae are exacerbated by increased body weight, a major feature of PCOS. The protein encoded by the *YAP1* gene plays a key role in one of the pivotal mechanisms that govern cellular/organismal metabolism and contributes to the pathogenesis of metabolic diseases. Aim: To compare the prevalence of single nucleotide variants (SNVs) in the *YAP1* gene among adolescents with PCOS, adolescents at risk of PCOS development and healthy adolescents, and assess their association with the clinical characteristics of PCOS. Results: The frequencies of the five investigated *YAP1* gene SNVs (rs11225161, rs11225166, rs3858420, rs11225138 and rs79981660) were not significantly different among adolescents with PCOS, risk group patients and healthy controls. Furthermore, none of the SNVs contributed to the clinical characteristics of adolescents with PCOS and adolescents at risk of PCOS development. Conclusions: No significant associations were found between PCOS in adolescents and the five investigated SNVs in the *YAP1* gene.

## 1. Introduction

Polycystic ovary syndrome (PCOS) is one of the most common endocrinopathies in women. It can manifest in adolescence and affects up to 8% of adolescents [1]. It is characterised by hyperandrogenism (biochemical and/or clinical) and oligomenorrhoea or amenorrhoea [2]. Adolescents with signs of PCOS but not meeting all the diagnostic criteria are considered a risk group and require additional testing around the gynaecological age of 8 years (chronological age minus menarche age), as a full diagnosis of PCOS can only be confirmed over time [2,3].

Approximately 50–70% of PCOS patients suffer from excess weight and adiposity, which can aggravate long-term health consequences characteristic of PCOS such as impaired fertility, increased risk of type 2 diabetes, metabolic disorders and cardiovascular disease [4,5,6,7].

Genome-wide association studies (GWAS) have identified numerous genetic loci that are linked to the development of PCOS and its clinical manifestations [8,9,10]. Several of these are associated with impaired metabolism, for example, *DENND1A*, *IRS-2*, etc. [8], and may contribute to the metabolic changes observed in PCOS patients.

One of the genetic loci linked to PCOS and identified through GWAS is located on chromosome 11q22.1 near the *YAP1* gene. Associations between PCOS and single nucleotide variants (SNVs) in the *YAP1* gene have been reproduced in several GWAS in different populations [8,9,10]. YAP1 protein plays a key role in the Salvador–Warts–Hippo (Hippo–Yap) signalling network that governs cellular and organismal metabolism. Deregulation of this pathway contributes to the pathogenesis of metabolic diseases such as type 2 diabetes, fatty liver disease and cardiovascular disease [11], all of which are potential long-term consequences of PCOS. Another PCOS-linked disease where YAP protein activity and its expression in cells is altered is endometrial cancer. Endometrial cancer development is also closely linked to insulin resistance—one of the hallmarks of PCOS [12]. Furthermore, a knockout mouse study found that YAP1 protein is required for the proliferation of ovarian granulosa cells [13], suggesting its modification may play a role in infertility, a major characteristic of PCOS patients.

To the best of our knowledge, only one genetic association study has investigated the possibility of a correlation between *YAP1* gene SNVs and the clinical characteristics of PCOS [14]. Additionally, characterizing protein–protein interactions in a discovery-validation GWAS, Zhang et al. found that both ERBB4 and WWTR1 can interact with YAP1 (all part of the Hippo–Yap signalling network) in PCOS patients, but no clinical correlation with different *YAP1* alleles was reported [15]. Further research focused on uncovering the role of *YAP1* in PCOS development and its clinical symptoms is clearly required.

The aim of this study was to determine the prevalence of carriers of pathogenic variants of the *YAP1* gene among adolescents with PCOS, adolescents at risk of PCOS development and healthy adolescents matched for gynaecological age. Moreover, we investigated whether there were any correlations between the *YAP1* gene SNVs and the clinical characteristics of PCOS.

## 2. Materials and Methods

### 2.1. Participants

Full study methodology is published elsewhere [16]. In brief: for this case–control study, we recruited adolescents at least a year after menarche, who attended the outpatient paediatric gynaecology clinic (Children’s Clinical University Hospital, Riga, Latvia). Cases were patients with an established diagnosis of PCOS according to the 2018 criteria of the European Society of Human Reproduction and Embryology (ESHRE) [2,16]. Control group participants were healthy adolescents who attended the same clinic due to reasons such as regular health care or contraception counselling. Patients with hirsutism but who did not fulfil all the PCOS criteria and did not have any exclusion criteria were included in the so-called ‘risk’ group, according to ESHRE Guidelines (2018), which states: ’For adolescents who have features of PCOS but do not meet diagnostic criteria, an “increased risk” could be considered and reassessment advised at or before full reproductive maturity, 8 years post menarche. This includes those with PCOS features before combined oral contraceptive pill (COCP) commencement, those with persisting features and those with significant weight gain in adolescence [2]. 

Gynaecological age was used for characterization of the age of the patient sample-age (in years) at the time of the study minus age at menarche, as it represents the phase of reproductive maturity in adolescence more accurately. 

World Health Organization AnthroPlus software was used to calculate BMI and its percentile according to age and normal range for adolescent girls (WHO: Geneva, Switzerland, 2009). Weights and heights were measured using standardized calibrated measuring devices.

For the sample size calculations, we used the following approach: study was prepared based on number of adolescents in Latvia in year 2018 (according to data from Central statistical Bureau)—in total, 52,408. Prevalence of PCOS in adolescent age group is 8% [1], confidence limit was set on 5%. Sample size for 80% confidence is 49 participants, for 90% confidence—80 participants. Calculation was performed in Open-source Epidemiologic Statistics for Public Health [17]. Statistical power was not calculated in this case. The exclusion criteria were serious comorbidities and use of hormonal medication within the previous six months. Recruitment took place between 1 January 2019 and 31 December 2020. Central Medical Ethics Committee of Latvia approval was obtained (protocol no. 1/16-04-12 and 3/21-02-17). 

### 2.2. Examination

The participants underwent thorough clinical examination, medical histories were obtained. Gynaecological ultrasound was performed by a single examiner using either an HD11 XE (Philips, Amsterdam, The Netherlands) or Logiq P5 (General Electric, Boston, MA, USA) ultrasound machine. The level of total testosterone was measured in the case group and the risk group patients on days 3–5 of their menstrual cycle (in a certified medical laboratory using an electrochemiluminescence method (Cobas 6000 immunological analyser; Roche, Basel, Switzerland)). Biochemical hyperandrogenism was defined as an elevated total testosterone level above the normal range on the Tanner scale [2].

### 2.3. SNV Selection

Three *YAP1* SNVs previously reported to be associated with PCOS were selected. Additionally, two other *YAP1* SNVs were selected from the gnomAD database (accessed on 17 February 2022) based on their frequencies being high enough to be detected in our patient group. None of the variants were in coding parts of the *YAP1* gene, as this type of variant is extremely rare. The five investigated SNVs and their PCOS associations are shown in Table 1. 

### 2.4. SNV Genotyping

Whole blood samples were obtained by peripheral venous puncture. Genomic DNA was isolated using the phenol-chloroform extraction method. Allelic discrimination of the SNVs rs11225161, rs11225166, rs3858420, rs11225138 and rs79981660 of the *YAP1* gene was performed using real-time PCR (TaqMan assays; Thermo Fisher Scientific, Waltham, MA, USA) using a modified manufacturer methodology. 

### 2.5. Statistical Analysis

IBM SPSS Statistics 22.0 version was used for the statistical calculations. The Mann–Whitney U test, Kruskal–Wallis H test and Fisher’s Exact test were used to evaluate the statistical significance of differences in median values or proportions of independent variables among the three study groups. For analysis of different inheritance models and *YAP1* SNV haplotypes, PLINK software was used.

## 3. Results

In total, 61 adolescent girls with PCOS, 22 risk group patients and 66 healthy adolescent girls participated in the study. The baseline clinical characteristics of the PCOS, risk and control group participants are presented in Table 2. The median age of PCOS patients was 16.0 (2.0), risk group and control group participants-15.0 (2.0) and 17.0 (1.0), respectively. In order to characterize the age of participants by group, we chose a more precise characteristic of the adolescent population—gynaecological age. Gynaecological age did not significantly differ among the groups (*p* = 0.258). Significantly more adolescents in the PCOS group and risk group showed markers of poor metabolic health: higher BMI, higher waist–hip ratio and higher percentage of participants with obesity (participants with BMI above 85th percentile for their age). Waist–hip ratio and BMI were significantly higher in PCOS and risk group patients than in control individuals (*p* < 0.001 and *p* = 0.001, respectively). In addition, BMI above the 85th percentile was significantly more common in individuals from the PCOS group than from the control and risk groups (*p* < 0.001). Overall, 34.4% of adolescent PCOS patients had characteristic polycystic ovary appearance on ultrasound examination, significantly more than in the risk and control groups (13.6% and 7.6%, respectively, *p* = 0.001). Prolactin, thyroid-stimulating hormone, oestradiol, luteinising hormone and follicle-stimulating hormone levels were in the normal laboratory range of premenopausal women (as per inclusion criteria).

The genotype frequencies of the five investigated *YAP1* gene SNVs in the three study groups are detailed in Table 3. For all five variants, there were no significant differences in the genotype frequencies among the PCOS, risk and control groups.

There were no significant differences in the allele carrier frequencies (including different inheritance models and haplotypes) among the PCOS, risk and control groups (data not shown).

None of the tested parameters (mFG score, BMI, waist–hip ratio, total testosterone level, luteinising hormone level, follicle-stimulating hormone level, oestradiol level and polycystic ovarian morphology on ultrasound) were significantly different among the PCOS patient genotypes.

The phenotype associations with homozygous major allele carriers (HH) and minor allele carriers (Hh and hh) in the PCOS group are shown in Table 4. *YAP1* rs11225166 minor allele carriers demonstrated a tendency towards a higher total testosterone level compared with homozygous major allele carriers; however, this result did not reach statistical significance (*p* = 0.08). Similarly, although statistical significance was not achieved (*p* = 0.09), *YAP1* rs11225161 minor allele carriers tended towards a higher follicle-stimulating hormone level compared with homozygous major allele carriers. Neither of these associations was found in the risk group.

## 4. Discussion

The frequencies of five *YAP1* gene SNVs (rs11225161, rs11225166, rs3858420, rs11225138 and rs79981660) were not significantly different among adolescents with PCOS, adolescents at risk of PCOS development and healthy controls. Interestingly, we found that the *YAP1* rs79981660 frequency in our study groups was different from that reported in Europeans in the gnomAD database (0.017–0.024 in our three groups versus 0.04). None of the tested parameters (mFG score, BMI, waist–hip ratio, total testosterone level, luteinising hormone level, follicle-stimulating hormone level, oestradiol level and polycystic ovarian morphology on ultrasound) were significantly different among the PCOS patient genotypes.

YAP1 is an intriguing protein with an established role in cellular metabolism and the pathophysiology of metabolic diseases. YAP1 protein upregulates glucose transporter activity, regulates insulin signalling and participates in amino acid metabolism and cell autophagy. It has a crucial role in lipid metabolism and consequently dysregulation of YAP1 protein function is involved in the pathology of non-alcoholic steatosis of the liver. It also contributes to atherosclerosis and cancer development through fatty acid metabolism [11]. All these conditions are hallmark metabolic consequences related to PCOS. In addition, YAP1 supports appropriate development and growth of ovarian follicles, while dysregulation in the Hippo pathway can lead to elevated expression of YAP1 and result in enlargement of the ovaries, which are characteristic to PCOS [18,19,20]. Because ovaries of PCOS patients appear fibrotic and YAP system is sensitive to mechanic stimuli, it seems likely that lower activity of Hippo signalling system elements observed in ovaries of PCOS patients, could contribute to an interaction between YAP1 and different tissue growth factors [21,22]. 

In the present study, no associations were found between the investigated *YAP1* gene SNVs and the development of PCOS and its clinical characteristics among adolescents with PCOS and risk group adolescents. Several GWAS studies have corroborated that the *YAP1* gene contributes to PCOS development [8,9,10]. SNV rs1894116 in *YAP1* gene has been reported to increase risk of PCOS development significantly (OR: 1.27, 95% CI: 1.20–1.36, *p* value: 1.08 × 10^−22^) [7,23]. There are only a few genetic association studies that have analysed *YAP1*’s contribution to the disease and its clinical characteristics. Li and colleagues’ study of a Han Chinese population reported that rs11225161 minor allele A carriers in the homozygote and heterozygote state (AA + AG genotypes) had higher blood glucose levels in oral glucose tolerance tests (OGTT). Furthermore, the rs11225166 minor allele (G) also contributed to higher blood glucose levels in OGTT. A further finding of their study was that the minor allele (G) of rs11225138 was a risk factor for higher luteinising hormone levels in PCOS patients. No other *YAP1* SNV/PCOS clinical characteristic associations were reported [14]. A possible explanation for the lack of a proven direct link between *YAP1* gene SNVs and PCOS-associated clinical characteristics could be that the Hippo–Yap signalling network has multiple inputs that may allow it to respond to subtle changes in multiple metabolites rather than a dramatic change in a single metabolite [11].

In the present study, we analysed *YAP1* gene SNVs, i.e., built-in changes in the DNA sequence in the blood of patients. However, there are other genetic mechanisms of *YAP1* gene functioning that could be involved in the development of PCOS and its clinical characteristics. Wang and colleagues found that YAP protein with other co-activators of the Hippo–Yap signalling network were related to insulin resistance in endometrial cancer cells from PCOS patients, with the effect being more pronounced in obese PCOS patients [12]. Additionally, epigenetic mechanisms that link the *YAP1* gene and PCOS have been proposed. Jiang and colleagues found that the methylation level of the *YAP1* promoter region in ovarian granulosa cells of PCOS patients was significantly lower than that of the control group. Thus, upregulation of YAP1 mRNA and protein expression levels was observed. They suggested that the testosterone concentration could alleviate the methylation status and contribute to higher expression levels of *YAP1*, which play a key role in the pathogenesis of PCOS and accelerate PCOS symptoms [24]. Another study analysed *YAP1* mRNA expression in granulosa cells of PCOS patients and healthy controls–they were upregulated together with mRNAs from other genes that are involved in follicular maturation *(EREG, ENTPD6)* [25]. 

Several authors reported that YAP1 could also play a role in the treatment of PCOS in the future. It is known that ovary damage promotes follicle growth in PCOS patients who underwent ovarian wedge resection or laparoscopic ovarian laser drilling. Furthermore, simply removing the ovarian cortex from these patients, followed by cutting of the tissue to disrupt ovarian Hippo signalling promoted follicle growth when cortical fragments were grafted back into patients. One can further develop less invasive therapies by directly injecting upstream Hippo signalling system elements, that activate YAP1 into the ovaries of these patients to obtain a higher number of mature oocytes prior to use of assisted reproductive technologies [20].

This study is a part of a broad research project. We have checked the association between multiple genetic variations in this sample (*GNRHR*, *ESR2*, *LHCGR*, *FSHR*, *CYP21A2*) and development of PCOS in adolescents so far. The genetic variations analysed were not associated with the development of PCOS in adolescents with PCOS. The incidence of these variations was also not higher in risk group patients than in healthy control group representatives. Adolescents with PCOS, who were carriers of the *ESR2* rs4986938 minor allele in homozygous state, and adolescents with the minor *LHCGR* rs2293275 allele in their genotype have higher total testosterone in blood than patients who are not carriers of these alleles [16]. More extensive haplotype analysis is planned in the future that incorporates results of this study with results of the previous studies. 

This study has several strengths. The calibre of the composition of our study population—rigorously examined adolescent PCOS patients, PCOS risk patients and age-matched healthy controls—was high. The mode of recruitment via free-of-charge gynaecologist consultations directly accessible at the national children’s hospital ensured that the three groups were similar socio-economically and demographically, thereby minimising the impact of other confounding variables. To the best of our knowledge, *YAP1* gene SNVs have not previously been studied in adolescent PCOS patients. Furthermore, as well as being the first study of its kind, our study used the latest ESHRE diagnostic criteria for PCOS in adolescents and also included a risk patient group [2]. A limitation of the study is its small sample size, which may have impacted the results. However, as all participants were subjected to strict diagnostic criteria and examination, we believe that our study results are applicable to a broader population. Adolescents with PCOS and at risk of PCOS development require close monitoring, and genetic testing is essential to understand the development of a particular phenotype and provide a prognosis for the future course of the disease. It should be noted that comparison of results from different research groups is difficult because of varying research methodologies and the updating of diagnostic criteria over the years.

## 5. Conclusions

No significant associations were found between PCOS in adolescents and the five investigated SNVs in the *YAP1* gene. Nonetheless, for two of the SNVs (rs11225166 and rs11225161), there was a tendency towards a higher testosterone and follicle-stimulating hormone level, respectively, in minor allele carriers.

## Figures and Tables

**Table 1 biomedicines-10-01688-t001:** Characterisation of selected SNVs in *YAP1* gene.

SNV	MAF in Europe ^a^	HGVS Nomenclature (Reference Sequence: NM_001130145.2)	Published Association [12]	Population
rs11225138	0.08	c.572+8773G > C	Higher LH level in PCOS group	Han Chinese, adults
rs11225161	0.08	c.803-6130C > T	Difference in PCOS and control group, additionally association with glucose level in OGTT	Han Chinese, adults
rs11225166	0.09	c.1033-3886G > T	Association with glucose level in OGTT	Han Chinese, adults
rs3858420	0.33	c.1276+42=	NR	NR
rs79981660	0.04	c.985-20G > A	NR	NR

^a^ Minor allele frequency (MAF) data from GnomAD (accessed on 17 February 2022) about European population. SNV—single nucleotide variation, HGVS—Human genome variation society, NR—not reported.

**Table 2 biomedicines-10-01688-t002:** Baseline clinical characteristics of the PCOS, risk and control groups.

Variable	PCOS Group (*n* = 61)	Risk Group (*n* = 22)	Control Group (*n* = 66)	*p*-Value
Chronological age, median years (IQR)	16 (2.0)	15 (2.0)	17 (1.0)	<0.001
Gynaecological age, median years (IQR) †	4.0 (3.0)	3.5 (2.0)	4.0 (2.0)	0.258
BMI, median percentile (IQR)	83.7 (50.4)	75.4 (42.1)	45.0 (46.6)	<0.001
Individuals with BMI above the 85th percentile, *n* (%)	30 (53.6)	6 (27.3)	9 (14.1)	<0.001
Waist–hip ratio, median (IQR)	0.82 (0.12)	0.80 (0.06)	0.76 (0.06)	0.001
mFG score, median (IQR)	9.0 (6.0)	8.0 (4.5)	2.0 (2.0)	<0.001
GAGS score, mean (SD)	14.4 (9.2)	10.9 (8.8)	6.9 (6.0)	<0.001
No acne, *n* (%)	1 (1.9)	2 (10.0)	6 (15.0)	
Mild acne, *n* (%)	34 (65.4)	13 (65.0)	31 (77.5)	0.149
Moderate acne, *n* (%)	15 (28.8)	4 (20.0)	3 (7.5)	0.003
Severe acne, *n* (%)	2 (3.8)	1 (5.0)	0 (0)	0.091
Polycystic ovary morphology on ultrasound, *n* (%)	21 (34.4)	3 (13.6)	5 (7.6)	0.001

† Gynaecological age—age (in years) at the time of the study minus age at menarche; BMI—body mass index; mFG—modified Ferriman-Gallwey; GAGS—Global Acne Grading System; IQR—interquartile range for medians.

**Table 3 biomedicines-10-01688-t003:** Genotype frequencies of the five SNVs in the PCOS, risk and control groups.

SNV/ Genotype ^c^	PCOS Group (*n* = 61)				Risk Group (*n* = 22)				Control Group (*n* = 66)			MAF	*p*-Value
	HH *n* (%)	Hh *n* (%)	hh *n* (%)	MAF ^a^	HH *n* (%)	Hh *n* (%)	hh *n* (%)	MAF ^b^	HH *n* (%)	Hh *n* (%)	hh *n* (%)	MAF	
*YAP1* rs79981660	57 (96.6)	2 (3.4)	0 (0)	0.017	20 (95.2)	1 (4.8)	0 (0)	0.024	60 (95.2)	3 (4.8)	0 (0)	0.024	1.00
*YAP1* rs11225138	52 (85.2)	8 (13.1)	1 (1.6)	0.082	18 (81.8)	4 (18.2)	0 (0)	0.091	56 (84.8)	10 (15.2)	0 (0)	0.076	0.81
*YAP1* rs11225161	49 (81.7)	10 (16.7)	1 (1.7)	0.100	15 (75.0)	5 (25.0)	0 (0)	0.125	49 (79.0)	13 (21.0)	0 (0)	0.105	0.71
*YAP1* rs11225166	46 (79.3)	11 (19.0)	1 (1.7)	0.112	17 (77.3)	5 (22.7)	0 (0)	0.114	47 (75.8)	15 (24.2)	0 (0)	0.121	0.83
*YAP1* rs3858420	27 (46.6)	29 (50.0)	2 (3.4)	0.284	11 (50.0)	7 (31.8)	4 (18.2)	0.341	35 (55.6)	22 (34.9)	6 (9.5)	0.270	0.14

HH—homozygous carriers of major alleles; Hh—heterozygous allele carriers; hh—homozygous carriers of minor alleles. ^a^ Genotypes for SNVS rs79981660, rs11225138, rs11225161, rs11225166, rs3858420 were available for 97%, 100%, 98%, 95%, 93% individuals, respectively. ^b^ Genotypes for SNVS rs79981660, rs11225138, rs11225161, rs11225166, rs3858420 were available for 95%, 100%, 91%, 100%, 100% individuals, respectively. ^c^ Genotypes for SNVS rs79981660, rs11225138, rs11225161, rs11225166, rs3858420 were available for 95%, 100%, 94%, 94%, 95% individuals, respectively.

**Table 4 biomedicines-10-01688-t004:** Phenotype associations with major or minor allele carrying in PCOS patients.

SNV	Modified Ferriman-Gallwey Score, Median (IQR)			BMI, Median Percentile (IQR)			Waist–Hip Ratio, Median (IQR)			Total Testosterone Level, Median (IQR)			LH, Mean (SD)			FSH, Median (IQR)			Estradiol, Median (IQR)			PCO Morphology on Ultrasound, *n* (%)		
	MAC (HH)	MiAC (Hh, hh)	*p*	MAC (HH)	MiAC (Hh, hh)	*p*	MAC (HH)	MiAC (Hh, hh)	*p*	MAC (HH)	MiAC (Hh, hh)	*p*	MAC (HH)	MiAC (Hh, hh)	*p*	MAC (HH)	MiAC (Hh, hh)	*p*	MAC (HH)	MiAC (Hh, hh)	*p*	MAC (HH)	MiAC (Hh, hh)	*p*
*YAP1* rs79981660	9.0 (6.0)	13.5	0.15	83.7 (51.5)	86.4	0.73	0.82 (0.12)	0.84	0.70	0.40 (0.40)	0.62	0.48	8.1 (5.1)	7.5 (5.6)	0.87	5.5 (2.5)	5.7	0.76	44.9 (24.9)	36.2	0.54	19 (35.8)	1 (50.0)	1.00
*YAP1* rs11225138	10.0 (6.0)	8.5 (5.8)	0.43	89.0 (56.6)	71.6 (33.3)	0.90	0.82 (0.11)	0.83 (0.25)	0.73	0.39 (0.40)	0.56 (0.38)	0.50	8.4 (5.2)	6.1 (4.4)	0.31	5.5 (2.5)	6.3 (4.1)	0.20	44.2 (23.2)	34.4 (27.1)	0.44	20 (40.8)	1 (12.5)	0.12
*YAP1* rs11225161	10.0 (6.0)	9.0 (6.0)	0.82	83.7 (58.0)	85.5 (33.0)	0.47	0.82 (0.11)	0.86 (0.22)	0.49	0.38 (0.41)	0.63 (0.36)	0.16	8.2 (5.2)	7.0 (4.9)	0.55	5.4 (2.5)	6.6 (2.8)	0.09	44.9 (23.7)	38.9 (20.9)	0.48	19 (41.3)	2 (20.0)	0.21
*YAP1* rs11225166	10.0 (6.0)	10.0 (5.5)	0.81	83.7 (58.0)	85.5 (33.0)	0.45	0.82 (0.11)	0.86 (0.22)	0.83	0.37 (0.40)	0.64 (0.38)	0.08	8.1 (5.1)	7.9 (5.3)	0.95	5.4 (2.4)	6.3 (2.5)	0.12	43.1 (23.2)	41.2 (29.5)	0.81	19 (43.2)	2 (18.2)	0.13
*YAP1* rs3858420	8.5 (6.5)	11.0 (5.8)	0.17	78.4 (64.0)	89.0 (47.0)	0.66	0.83 (0.11)	0.82 (0.13)	0.42	0.41 (0.42)	0.41 (0.38)	0.89	7.4 (4.9)	8.6 (5.3)	0.46	5.5 (2.8)	5.6 (2.2)	0.25	45.0 (32.2)	37.5 (21.0)	0.14	11 (44.0)	10 (33.3)	0.42

PCO—polycystic ovary; MAC—major allele carrier; MiAC—Minor allele carrier; HH—homozygous carriers of major alleles; Hh—heterozygous allele carriers; hh—homozygous carriers of minor alleles; IQR—interquartile range for medians.

## Data Availability

Not applicable.

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
