# Peer review of "Role of Single Nucleotide Variants in the YAP1 Gene in Adolescents with Polycystic Ovary Syndrome"

_biomedicines, 2022, doi:10.3390/biomedicines10071688_

Round 1

Reviewer 1 Report

In perspective of already existing evidence the rationale of the study was justified. Moreover, concise and short introduction presents interactions and associations of YAP1. I have rather technical sugestions than substantials remarks. 

1. I'm not convinced about prevalence of PCOS, I would say one of the most common endocrinopathy, in line with Hashimoto... 

2. I understand that methodology was similar to other, already presented study. It is ok. However, I would suspect that initially the reader will get info about quantities in the groups. Moreover, as I verified, you analyzed other parameters on such study model. Why you didn't mention about this in the present manuscript and didn't introduce those results into the present statistics (mentioning about this)? 

3. One of the most dissapointing parts of the paper are references... The is planty of interesting and important manuscripts which are up-to-date and worth to be mentioned. What about current diagnostic and therapeutic recommendations? Reviews on PCOS? I.a. 

10.1210/endrev/bnac013
10.3389/fmed.2021.741803
10.1007/s43032-020-00375-4

It may also have visible impact on the more comprehensive discussion. 

Author Response

We are very greatful for your thorough revision and tried to accomodate all your suggestions. 1) ‘’The most common’’ is changed to one of the most common; 2) Number of participants in the research groups are indicated in the result section; 3) We included short summary of the previous research conducted by this research group in discussion section. We hope to be able to do haplotype analysis (that includes other SNV’s analyzed) in the future. Unfortunately, as sample size is quite small in this phase of the study, preliminary haplotype analysis did not reveal any significant results. Several haplotype groups where with extremely small number of individuals. 4) Thank you for your valuable reference suggestions. We included those and some other as well.

Reviewer 2 Report

Abstract

Please delete ‘Nonetheless, for two of the SNVs 33 (rs11225166 and rs11225161), there was a tendency towards a higher testosterone and follicle-stim-34 ulating hormone level, respectively, in minor allele carriers.’ in Conclusion.

Introduction

1.       Please add the references for ‘long-term health consequences characteristic of PCOS such as im-47 paired fertility, increased risk of type-2 diabetes, metabolic disorders and cardiovascular’

-1. Gorry, A., White, D. M., & Franks, S. (2006). Infertility in polycystic ovary syndrome. Endocrine30(1), 27-33.

2. Ovalle, F., & Azziz, R. (2002). Insulin resistance, polycystic ovary syndrome, and type 2 diabetes mellitus. Fertility and sterility77(6), 1095-1105.

3. Karaer, A., Cavkaytar, S., Mert, I., Buyukkagnici, U., & Batioglu, S. (2010). Cardiovascular risk factors in polycystic ovary syndrome. Journal of Obstetrics and Gynaecology30(4), 387-392.

Materials Methods

1.       Please describe the adolescents at risk of PCOS development. What is the characteristics of risk group

2.       Please add the number of study participants in each study group

3.       Please add the mean age of study groups.

4.       Sample size calculation and power analysis must be done.

Results

1.       The significant p-value is not shown in bold.Please delete the sentence

2.       Please don’t mentioned that ‘a higher total testosterone level compared with homozygous ma-184 jor allele carriers; however, this result did not reach statistical significance (p=0.08).’

Author Response

Thank you for your comments. We hope we have managed those successfully. 1) The sentence is deleted from conclusion section and non-significant result regarding testosterone level excluded from the article. 2) Complimentary references are included in the article; 3) We added a bit more information regarding the risk group throughout the article; 4) Number of participants in each group indicated; 5) Mean age of study groups added; 6) Comment regarding sample size calculation added;

Round 2

Reviewer 2 Report

The article is improved after revision.

Patients with hirsutism but who  did not fulfil all the PCOS criteria and did not have any exclusion criteria were included in the so-called ‘risk’ group, according to ESHRE Guidelines (2018) [2]. Please add more detailed information about 'risk group' definition? 

How many of the risk group were diagnosed with PCOS as a result of follow-up?

Please add the mean+SD age in Table. Please add one sentence about the gynecologic age in Methods.

What is  BMI, median percantile? and above the 85th percentile? Please delete these lines in Tables ( and in Result section). Instead of please add the mean+ SD or median (IQR) BMI value in Table?

Author Response

Dear reviewer

Thank you for your comments.

  • According to ESHRE, 2018. guidelines, the exact definition of the risk group is stated in 1.1.3. recommendation. Verbatim text: ‘’For adolescents who have features of PCOS but do not meet diagnostic criteria, an “increased risk” could be considered and reassessment advised at or before full reproductive maturity, 8 years post menarche. This includes those with PCOS features before combined oral contraceptive pill (COCP) commencement, those with persisting features and those with significant weight gain in adolescence.’’ This also answers to you second remark – at minimum 8 year follow-up period should be observed after the beginning of the menstruations. It means that follow-up for this sample will be carried out, but we still need to wait couple of years for the participants to reach the appropriate age, till this moment these patients are ,,risk group’’. In order to avoid any confusion, we added the exact text in the study paper.
  • We added the chronological age at the Table and explanation regarding gynecological age in methods. We hope it clarifies the information.
  • According to WHO guidelines, BMI percentile is used for adolescent age group, rather than BMI value. As there are different cut-off values for normal weight, overweight and obesity for the age of the patient. 85th percentile is a cut-off point, that marks increased weight. For each participant this was calculated by using WHO AnthroPlus programme. We added explanation in Methods section and results.

We will be happy to provide any additional corrections.

Best regards,

Lasma Lidaka

Round 3

Reviewer 2 Report

The article is suitable for publication in Biomedicines.

Yours sincerely,